# Automotive Augmented Reality Head-Up Displays

**DOI:** 10.3390/mi15040442

**Published:** 2024-03-26

**Authors:** Chen Zhou, Wen Qiao, Jianyu Hua, Linsen Chen

**Affiliations:** 1School of Optoelectronic Science and Engineering & Collaborative Innovation Center of Suzhou Nano Science and Technology, Soochow University, Suzhou 215006, China; 20235239035@stu.suda.edu.cn (C.Z.); jyhua@suda.edu.cn (J.H.); lschen@suda.edu.cn (L.C.); 2Key Lab of Advanced Optical Manufacturing Technologies of Jiangsu Province & Key Lab of Modern Optical Technologies of Education Ministry of China, Soochow University, Suzhou 215006, China

**Keywords:** heads-up display, augmented reality, picture generation unit, optical design

## Abstract

As the next generation of in-vehicle intelligent platforms, the augmented reality heads-up display (AR-HUD) has a huge information interaction capacity, can provide drivers with auxiliary driving information, avoid the distractions caused by the lower head during the driving process, and greatly improve driving safety. However, AR-HUD systems still face great challenges in the realization of multi-plane full-color display, and they cannot truly achieve the integration of virtual information and real road conditions. To overcome these problems, many new devices and materials have been applied to AR-HUDs, and many novel systems have been developed. This study first reviews some key metrics of HUDs, investigates the structures of various picture generation units (PGUs), and finally focuses on the development status of AR-HUDs, analyzes the advantages and disadvantages of existing technologies, and points out the future research directions for AR-HUDs.

## 1. Introduction to Heads-Up Displays (HUDs)

Traffic crashes rank as the sixth leading cause of disability-adjusted life-years lost globally, standing out as the sole non-disease factor among the top 15 contributors according to a report by the World Health Organization (WHO) [1]. The WHO’s findings from 2018 revealed that approximately 1.3 million individuals succumb annually due to road traffic accidents [2]. Moreover, some 94% of them are mainly contributed by human error [3,4]. Specifically, people behind the wheel are exposed to various of factors that may lead to distractions, such as navigation aids and instrument panels. The road environment is unpredictable, and when a driver’s eyes switch back and forth between the vehicle information inside the car and the real-time road conditions outside the car at a distance, it is very easy to cause traffic accidents [5]. As a perfect solution, a heads-up display (HUD) is an interaction-based in-vehicle display technology that projects driving information onto the physical scene beyond the windshield and improves driving safety [6,7]. The driving information includes speed, fuel consumption, navigation data, driver assistance information, warning messages, etc. In addition, an HUD can be combined with autonomous driving to provide a better human–machine interaction experience.

Since the primitive concept was applied for aiming and shooting in 1960, the field of HUDs has made significant advancements [8], especially with the emergence of augmented reality heads-up displays (AR-HUDs) in recent years, which has enabled virtual images to be superimposed on traffic environments because of the large virtual image distance (VID) [9]. So, this avoids a frequent change in the accommodation of the human eye between the physical world and the displayed information [10]. In terms of optical hardware, an HUD is composed of a picture generation unit (PGU) and optics for the HUD. The former utilizes projectors, such as a thin-film transistor–liquid crystal display (TFT-LCD), digital light processing (DLP), liquid crystal on silicon (LCOS), or micro-LEDs, to generate images. The latter is an optical system for projecting a virtual image beyond the windshield. Its function includes folding the light path, magnifying the image, forming an eyebox, and compensating the aberration from the windshield (Figure 1).

This review focuses on the research advances in HUD optical systems. Following the introduction, we first introduce the key metrics in HUDs and the technical progression of commercial HUDs. Next, we introduce the PGUs that are generally used in all kinds of HUDs. We further categorize the optics of HUDs according to the features of the virtual image plane. The fundamentals, display performance, and challenges are discussed. Finally, we will explore the major obstacles of AR-HUDs and highlight the potential future directions.

## 2. Metrics of HUDs

An AR-HUD is a product that combines many fields, such as optical displays, LIDAR detection [11], data processing [12], etc. Although there are numerous parameters for evaluating such systems, this study focuses only on the optical architecture and introduces the metrics of optical-system-related parameters, including the field of view (FOV), virtual image distance, eyebox, and volume of the system [13,14,15] (Figure 2).

The FOV refers to the observable angle range of a virtual image. In an ideal HUD, the virtual image should cover a minimum of two lanes (the vehicle’s driving lane and half a lane on each side) for effective traffic illustration. Considering a road width of approximately 3.5 m, the minimum horizontal FOV of an HUD should be 20° [16]. Unfortunately, the horizontal FOV of a typical AR-HUD is 10°.

The VID represents the distance between the virtual image and the human eye. To effectively integrate navigation information with the road scene, it is recommended that the VID should exceed 10 m, ideally surpassing 20 m [17]. Increasing the virtual image distance not only enlarges the projection size but also enhances interactivity by enabling a greater amount of information to be presented to the driver.

The interocular distance between human eyes is approximately 65 mm. To ensure the inclusion of both eyes within the eyebox, it is a common practice to set an eyebox size larger than 90 × 60 mm^2^ [18]. In order to prevent the loss of driving information during vehicle swaying, it is advantageous to have a larger eyebox. In addition, a large eyebox allows drivers to slightly adjust their driving position, thereby significantly enhancing the overall driving experience.

In order to meet the application requirements, the size of the whole HUD system should be as small as possible. A compact system is always the goal pursued by many researchers. Each of the aforementioned performance metrics is not independent; instead, they exhibit a contradictory yet unifying relationship among them. For example, the system volume usually increases as the FOV increases. The objective for researchers lies in striking a harmonious balance among these diverse indicators.

HUDs have undergone three rounds of technological advancement: combiner HUDs (C-HUD), windshield HUDs (W-HUD), and AR-HUDs [19,20,21,22]. C-HUDs are based on a separate optical panel placed above the dashboard, so the HUD’s optical system can be designed and optimized independently of the profile of the windshield, with a lower cost and system complexity. Due to the width limit of the standalone panel, the display distance is small (~1 m) [23]. More importantly, since the display is located inside the vehicle, the driver needs to switch his/her focus away from the outside world. Because of the abovementioned drawbacks, W-HUDs and AR-HUDs have been developed. The windshield is used as a combiner to display the projected information. As a result, the FOV and VID are greatly improved compared with those of C-HUDs. AR-HUDs have a larger FOV and VID than those of W-HUDs, and they can integrate navigation information, warning information, entertainment information, etc. with the real road, realizing human–machine interaction and greatly improving driving safety, and they have become the main development direction for future in-vehicle heads-up display systems. Table 1 summarizes the critical parameters of the three HUD at different stages of development.

## 3. Picture Generation Unit for HUDs

Currently, commercial HUDs mainly employ four kinds of projection technologies: LCD-TFT [26], DLP [27], LCOS [28], and Micro-LEDs. Among them, LCOS can be categorized into two types: the amplitude type and phase type [29]. In the following, we will begin with the principle of picture generation units, based on which the characteristics of different types will be analyzed.

The principle of the LCD-TFT is based on the characteristic that the voltage-induced liquid crystal can modulate the polarization direction of incoming linearly polarized lights. To be specific, when light passes through the liquid crystal layer, it splits into two beams. Due to their different speeds but the fact that they have the same phase, when two beams are recombined, there will be an inevitable change in the vibration direction of the resulting light [30]. As shown in Figure 3a, when light passes through the lower polarizer, it becomes linearly polarized with a vibration direction consistent with that of the polarizer, and the liquid crystal has a specific rotation direction without applying voltage [31]. Consequently, as light traverses through the liquid crystal layer, gradual distortion occurs until reaching the upper polarizer, where its vibration direction is rotated by 90° and aligns with that of the upper polarizer. Upon applying voltage, electric-field-induced orientation eliminates this distortion effect within the liquid crystal layer. Linearly polarized light no longer rotates within this layer so that it cannot traverse through the upper polarizer to form a bright field. By combining liquid crystal with a color filter, an LCD-TFT can be used to realize full-color display, and this is the most common transmission projector.

Unlike the LCD-TFT, DLP technology utilizes a digital micromirror device (DMD) to steer incident light. A DMD is a complex optical switching device consisting of 1.3 million hinged micromirrors, one for each pixel in a rectangular array. The hinged micromirror can reflect incident light in two opposite directions corresponding to the on and off states. So, with pulse width modulation, DMD can generate multiple gray levels. As shown in Figure 3b, DLP generates full-color images through field sequential color operation. According to this principle, the incident light does not need to be polarized; thus, a higher optical efficiency compared with that of LCD-TFT and LCOS is achieved.

LCOS also employs the photoelectric effect of liquid crystals to modulate the incident linearly polarized light (Figure 3c). However, unlike TFT-LCD, LCOS utilizes a complementary metal-oxide semiconductor (CMOS) as the lower substrate, which is employed as a reflector that allows for the integration of LCOS transistors and lines within a CMOS chip (lower substrate) positioned beneath the reflective surface. This integration optimizes surface area utilization and results in a greater opening rate. As a result, LCOS achieves higher resolution and brightness and mitigates pixelation artifacts. In addition, LCOS can also be used as a phase modulation device in the field of holography [32]. By loading holograms onto a phase-type LCOS, it becomes possible to reconstruct images with depth information under coherent light illumination. As a result, an LCOS with phase modulation may be used in an AR-HUD to achieve 3D effects in the near future.

Recently, inorganic-material-based light-emitting diodes (LEDs) with sizes down to less than 50 μm (which are also called micro-LEDs) have attained much more attention for display technologies [33]. Each pixel can be addressed and driven to emit light individually, as shown in Figure 3d. Therefore, micro-LEDs distinguish themselves due to their small power consumption, low contrast ratio, and high resolution. During their manufacturing, the epitaxial growth of InGaN blue and AlInGaP red micro-LEDs requires different substrates. Subsequently, these micro-LED chips need to be transferred onto a silicon backplane with specific pixel spacing. However, the large-scale transfer technology involved in this process remains a challenge [34,35]. The approach of using only blue micro-LEDs with extremely high pixel density to excite quantum dots (QDs) in a color conversion array to realize full-color display is a feasible solution for easing their manufacturing [36,37]. When the QD material as the color conversion layer is exposed to heat, light, or humidity, its stability becomes an issue [38]. Although micro-LEDs currently still demand substantial development efforts, it is believed that with the development of chip technology, their follow-up potential is huge [39].

Table 2 summarizes the critical parameters of the three aforementioned PGUs. Both the LCD-TFT and amplitude-type LCOS technologies utilize the photoelectric effect of liquid crystals to modulate the polarization state of incident light in combination with a polarizer for the purpose of adjusting light intensity. As a result, the light emitted from both technologies is linearly polarized. Although the light efficiency is dramatically reduced because of the polarization state at the device level, the optical efficiency might be comparable with that of a DLP-based HUD at the system level because of the polarization-sensitive reflectance of the windshield. High image contrast is desirable in HUD systems to eliminate the “darkness window”. DLP generally maintains a better image contrast. It is hard for the LCD-TFT and LCOS technologies to enhance the contrast ratio due to the persistent backlighting. To address this issue, local dimming techniques can be employed. On the other hand, the high cost of DLP technology poses a significant barrier to its widespread application.

## 4. Optics for HUDs

In HUD systems, PGUs with different characteristics need to be combined with purposely designed optics to achieve satisfactory display effects. For AR-HUDs, the distance of the virtual image plane is a vital parameter for evaluating the display effect, and it directly determines whether the image can be integrated with the real scene. To be specific, a system with only one image plane needs to push the VID to around 10 m from the observer so that the virtual image can fuse well with the real driving environment. Furthermore, an HUD with a multi-image plane or even a 3D image plane can provide more auxiliary information at different depth planes based on objects at different distances on the road. In the following, we classify AR-HUDs according to their different virtual image planes and introduce the relevant state-of-the-art research progress.

### 4.1. Virtual Image Distance within 10 m

HUDs based on off-axis reflection have been widely commercialized thanks to the mature manufacturing process of geometric optics. Nevertheless, their image planes tend to be close—typically below 8 m—and they have a large volume. A common design for AR-HUD systems is the free-space-based off-axis two-mirror system. The system, as shown in Figure 4a, utilizes a first fixed free-form surface as an optical path folder to reduce the system’s volume. The second free-form surface reflects the enlarged image onto the windshield, allowing the driver to see a magnified image in the distance. The advantage of using free-form mirrors in optical AR-HUD systems is that they are free of chromatic aberration [40] and can help eliminate off-axis aberrations [41]. Free-form surfaces not only correct image aberrations but also enable the system to be relatively compact in size (10L+).

However, the optical structure described above presents a trade-off among the FOV, VID, eyebox size, and system volume. A larger FOV and farther VID require free-form surfaces to be enlarged, which poses a challenge in maintaining a compact system volume. In 2020, an off-axis four-mirror system was proposed to meet the requirement of an adjustable exit pupil height while maintaining compactness [42]. This system had an eyebox size of 106 × 66 mm^2^, an FOV of 6° × 3°, and a VID of 5 m. As shown in Figure 4b, the four-mirror system consists of four reflection surfaces (labeled as 1, 2, 3, and 4) with a combiner. The light emitted from the GPU undergoes multiple reflections and eventually reaches the combiner, which reflects the light toward the exit of the system. The combiner allows observers to perceive both virtual images and reality simultaneously. Spherical surface 1 can be replaced with a free-form surface, and spherical surfaces 2 and 4 can be replaced with aspherical surfaces to balance non-symmetrical aberrations.

While free-space-based optics offer cost-effectiveness and simplicity, the system volume tends to become larger, bulkier, and more difficult to compactly integrate as the desired FOV, VID, and eyebox size for AR-HUDs increase [43]. Limited to a close image distance (usually smaller than 7 m), the windshield will produce a ghost image that cannot be ignored. Specifically, when a beam of light is incident on the first and second surfaces of the windshield at an oblique angle, the two reflected light rays will not completely coincide, and they will reach the human eye in a staggered form (Figure 5a) [44,45,46]. To reduce ghost images, a PVB film in the two glass layers on the windshield is modified into a wedge; from the traditional equal thickness, it is changed to be thick on the top and thin on the bottom, showing a wedge angle. After the light refracted through the first surface of the glass touches the second surface of the windshield glass, the reflection height and angle also change, making the two virtual images formed by the incident light on the first and second surfaces coincide, thus reducing the phenomenon of double shadow [47]. Moreover, utilizing a partial transparent Fresnel reflector as the combiner could also eliminate ghost images with the Fresnel pattern [48], since it can prevent a ghost image from reaching the eyes of drivers (Figure 5b).

In contrast, waveguide optics with ultra-thin flat optical waveguides and a far-image plane significantly reduce system volume. In addition, when the virtual image distance is far enough, the impact of ghost images is almost negligible. This makes waveguide optics an important direction for the future development of AR-HUDs with great market value and prospects [49].

### 4.2. VID Larger Than 10 m

Indeed, a truly future-proof AR-HUD must possess a large FOV (>10° × 5°), an extended VID (>10 m), and a compact form factor. While existing free-space optical solutions can already realize part of the AR-HUD functionality, the attempt to further enhance the system parameters would result in a significantly large volume. Consequently, waveguide schemes are desirable because of their minimal impact on thickness with varying FOVs and the ability to create an image at an infinite distance (>15 m), thereby aiming to achieve a superior AR-HUD.

In the waveguide scheme, the light emitted from the projector is coupled into the waveguide medium and propagated to the decoupled region through total internal reflection (TIR). Subsequently, multiple decoupling events occur in the decoupled region, leading to an enlarged effective eyebox size. This approach enables a greater utilization of the waveguide area while maintaining a smaller system volume (2V+). Geometric waveguides and diffraction waveguides can be distinguished based on the coupling methods.

Geometric waveguides employ partially reflective mirrors as the simplest elements for coupling in and out. As depicted in Figure 6a, a fully reflective mirror serves as the coupled-in structure, reflecting light from the projector into the waveguide [50]. Cascaded embedded partially reflective mirrors (PRMs) function as out-couplers, extending the exit pupil [51]. Figure 6b shows that a prism array can also serve as either an in-coupler or out-coupler in a waveguide. However, due to the multiple reflections of rays on the PRMs, issues such as stray light [52], ghosting, and low beam energy uniformity may arise [53,54].

Diffractive couplers modulate the light field through diffraction. Among the various diffractive structures, gratings, such as surface-relief gratings (SRGs) and volume holographic gratings (VHGs), are most commonly employed.

As shown in Figure 6c, one-dimensional SRGs modulate the phase distribution of incident light through a periodically undulating surface structure to diffract light into or out of the waveguides. When the surface relief grating modulates the beam, its transmission strictly follows the diffraction equation of light, and the diffraction direction is determined by the wavelength and incidence angle of the incident light, the grating period, and the refractive index of the material. One-dimensional gratings can be categorized into rectangular, trapezoidal, blazed, and inclined gratings based on their undulating shape. To reduce zeroth-order diffraction and increase optical efficiency, blazed gratings [55] and multilevel structures [56] are commonly employed in SRGs. By adjusting the diffraction efficiency of the diffraction gratings in different regions, the system can manipulate the beam energy coupling from the waveguide, allowing a uniform distribution of beam energy within the expanded eye box. However, due to the relatively low diffractive efficiency of SRGs, a PGU with high brightness is necessary, which increases the system’s energy consumption.

VHGs are composed of holographic optical elements (HOEs) created by illuminating a photosensitive film with two beams of coherent waves [57]. They transfer the intensity information of the interfering lights into transmittance modulation. VHGs offer advantages such as high resolution, low cost, low scattering, and simplicity in fabrication (Figure 6d). In 2019, C. M. Bigler et al. proposed a holographic waveguide HUD system with two-dimensional pupil expansion using VHGs [58]. The work achieved an FOV of 24° × 12.6°, expansion of the horizontal exit pupil by 1.9 times, and expansion of the vertical exit pupil by 1.6 times at an observation distance of 114 mm.

Polarization volume gratings (PVGs) are also extensively studied in the waveguides of augmented reality glasses, serving as holographic optical elements that record the interference between right-handed circularly polarized (RCP) and left-handed circularly polarized (LCP) beams [59,60,61]. PVGs offer the advantage of achieving high diffractive efficiency through the utilization of Bragg structures. However, the successful integration of PVGs into HUDs necessitates addressing the challenge of ensuring consistent performance in large-scale production. Failure to address this issue could lead to variations in image quality, color fidelity, or other visual artifacts, compromising the overall user experience and effectiveness of the HUD.

There is a dispersion problem in the realization of full-color display for diffractive waveguides because of the wavelength selectivity (Figure 7a). For SRGs, the most commonly used solution is to separately process beams of different bands by using multi-layer waveguides [62] (Figure 7b). However, a multi-layer waveguide will undoubtedly increase the volume and weight of a system. For VHGs, a method to avoid the issue caused by multiple waveguides is to stack multi-layer VHGs together (Figure 7c). An alternative solution is to use spatial multiplexing [63], since its most unique property is that several volume gratings are recorded in a single material. Moreover, multilayered plasmonic metasurfaces are adopted as achromatic diffractive grating couplers [64] with the features of an ultra-thin (<500 nm) form factor and high transparency (~90%). Three layers of such metasurface gratings are stacked together to couple a full-color image into a single waveguide.

In conclusion, the advantages of geometric waveguides include a large field of view, high light efficiency, and low color dispersion [65], but there are still difficulties in their large-scale processing and manufacturing. When machining geometric optical waveguides, engineers cut glasses according to the specified angle, polish the cutting surface, coat the surface with reflective film, and then glue several pieces of glass. In order to ensure the final image quality, the parallelism tolerance between different PRMSs in the waveguide structure is very critical, leading to a relatively low yield.

Diffractive waveguides can be processed through nano-imprinting or tow-beam interference. These two processing schemes are scalable, so the mass production of large-size diffractive waveguides is feasible. As a result, waveguide-based AR-HUDs are expected to be one of the most possible solutions for a small volume and large FOV [66].

### 4.3. Multi-Image Distance

Multiple virtual image depth planes or the 3D depth of field are another important optical metric for HUDs. Intuitively, at least two virtual depth planes are required: a near-depth plane at 2–5 m to present driving status information, such as speed, fuel level, etc., and a virtual plane at 10 m or more to present navigation information. The organic separation of virtual information improves the effectiveness and smoothness of information presentation. Moreover, the natural scenery outside a vehicle has different depths. An ideal HUD should be able to provide virtual information that matches the depth and distance of natural objects. However, the current HUD provides only one virtual image plane [67,68,69], which limits the fusion of virtuality with reality [70]. Therefore, multiple virtual image plane, depth-variable virtual image planes, and real 3D virtual scenery have become the main research directions for next-generation heads-up displays.

Two image planes can be achieved by employing two PGUs in a dual optical route separation design at the expense of an enlarged system volume [71]. Recently, an HUD with dual-layer display functionality that used a coupling architecture of near-field and far-field dual optical routes was proposed [72]. The near-field image plane was set to 2 m, and the far one was set to a distance of 8–24 m. The FOVs were 6° × 2° and 10° × 3°, respectively. As shown in Figure 8a, three folding mirrors were employed to fold the far-field optical path and connect it with the near-field optical path. The single-PGU approach for linking near-field and far-field dual optical routes can reduce the system volume, but it was achieved at the expense of reduced spatial resolution of the virtual image. In addition, the focus-free characteristic of a laser source can also be used to implement two images on different screens with a single PGU. A multi-depth heads-up display system that uses a single-laser-scanning PGU was proposed by Seo et al. [73]. The system can project an image to distances of 2 m and 5 m (Figure 8b). The projection distance and magnification of the virtual image are determined by the surface profile of the aspheric mirror and the throw distance.

The abovementioned multi-depth image is generated with different objective distances in the optical path. Alternatively, diffractive optical elements can also provide multiple depths. For example, using a combination of a hololens and holographic grating, two virtual images can be realized near and far [74]. The combination of hololens and hololeveling devices provided a multi-plane AR-HUD with dual modes of real–virtual images (Figure 8c) [75]. As shown in Figure 8d,e, the distance of the virtual image is 5 m, while the field-of-view angles for the real and virtual images are 10° × 4° and 8° × 4°, respectively.

**Figure 8 micromachines-15-00442-f008:**
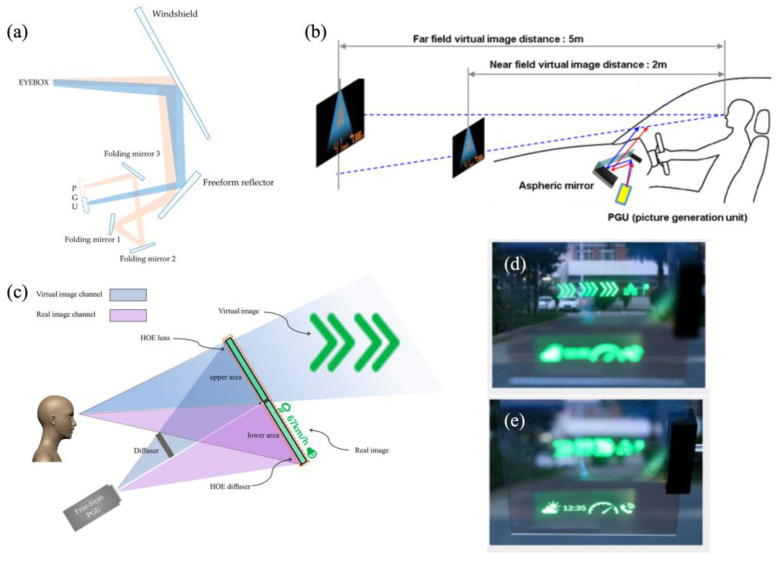
Research progress for multiple-image-plane HUD systems based on a single PGU. (**a**) Schematic diagram of a multiplane HUD system based on LCD-TFT [72], copyright of Jiang. (**b**) Schematic diagram of a multiplane HUD system based on LBS [73], copyright of Seo. (**c**) Schematic diagram of a multiplane HUD system based on HOE. (**d**,**e**) Dual image planes on a road [75], copyright of Optica.

### 4.4. Variable Image Distance

Aside from multi-depth HUD systems, the fusion of virtuality and reality can be improved by projecting images to variable depths [76,77]. In 2020, Kun et al. presented an HUD with a dynamic depth-variable viewing effect based on a liquid lens [78] (Figure 9a). The system comprises a liquid lens (L-Lens), an achromatic lens (A-Lens), and a Fresnel lens (F-Lens). The VID varies from 1.52 m to 1.95 m when the optical power of the L-Lens changes from −4.5 to 5.5 diopters (Figure 9b,c), but the VIDs of both planes are below 2 m. Nevertheless, currently, there is a limited amount of research on the aberration of the tuning lens, and improvements are needed to enhance the optical performance [79,80,81]. Furthermore, the tuning range is limited, and further investigation into focusing hysteresis phenomena is necessary [82]. The Pancharatnam–Berry optical element (PBOE) exhibits exceptional phase modulation capabilities for polarized light. Recently, Tao et al. proposed an HUD with Pancharatnam–Berry lenses (PBLs) to effectively switch the depth of virtual images [83]. As shown in Figure 9d, the PBL was set between the mirror and the combiner to alter the virtual image depth by adjusting the input polarization states to accommodate diverse driving scenarios.

Utilizing dynamically adjustable spatial light modulators, the projected images’ position can be modified without any mechanical movements. Mu et al. integrated a phase-type LCoS to continuously adjust the depths of holographic images within the range of 3 m to 30 m [84]. The display image size, resolution, and field of view are constrained by a state-of-the-art SLM.

To sum up, compared to the traditional multi-optical off-axis reflection scheme, HUDs based on novel photonic devices (such as the PBOE and SLM) replace the system complexity with material and manufacturing intricacies of a single optical component. However, during the transition from cutting-edge research to industrial production, factors such as material stability and optical device manufacturing costs still necessitate meticulous consideration.

### 4.5. Three-Dimensional HUDs

Two-dimensional displays with multiple depths or variable depth rely on algorithms for the postprocessing of a few 2D images, such as through coordinate conversion, to achieve 3D visual depth, but they are still limited in the continuous depth reconstruction. Therefore, 3D-HUDs represent an inevitable trend in the development of heads-up displays by combining heads-up displays with 3D displays. A 3D-HUD promptly updates the vehicle status and projects various information reflecting the vehicle’s driving condition to different depths of field, facilitating prompt reaction by drivers based on dynamic information from both virtuality and reality.

Takuya et al. proposed a 3D AR-HUD system based on a parallax barrier and eye tracking [85]. As depicted in Figure 10a, a minimum crosstalk of 2.08% and a 3D depth range of 1–20 m were achieved at the optimum viewing distance (OVD). To expand the FOV and provide horizontal parallax, Anastasiia et al. presented an interesting 3D-HUD combined with a waveguide for pupil replication with two lens arrays [86]. An eye-tracking system, a PGU, and a spatial mask were employed and synchronized to generate two eyeboxes for binocular vision through time multiplexing. The viewing angle was expanded from 12.5° × 7° to 20° × 7° by the telescopic architecture.

HOEs are crucial photonic devices in augmented reality display and transparent display, effectively manipulating light fields as lenses, gratings, or diffusers with high spectral and angular selectivity. At the same time, HOE-based combiners are the some of the most promising photonic devices for achieving a full windscreen HUD. Most recently, Zhen et al. proposed a 3D heads-up display that integrates a microlens-array-based 3D display module with a holographic combiner [87] (Figure 10b). By selectively recording varying optical powers at different locations of the HOE combiner, the holographic combiner achieves image amplification and off-axis diffraction. As shown in Figure 10c,d a depth range from 250 mm to 850 mm corresponding to green and red was achieved in their experiment.

In general, the aforementioned 3D-HUD schemes integrate existing 3D display technology (such as 3D displays based on parallax barriers, column lens arrays, spatial–temporal multiplexing, vector light fields, etc. [88,89,90,91,92,93,94,95]) with an eye-tracking algorithm to generate separate eyeboxes for both eyes in order to minimize crosstalk. However, due to the vergence–accommodation conflict, it is still under investigation whether discomfort will occur for high-speed vehicle drivers.

A holographic 3D display [96,97] reconstructs a light field by using an SLM dynamic loading hologram. Therefore, a holographic projection scheme can be integrated into an HUD [98,99]. In the field of computer-generated holography (CGH) algorithms, CNN-assisted CGH has made tremendous progress during the last few years [100,101]. However, limitations exist from the perspective of hardware implementation. For example, the presence of speckle noise caused by the interference and diffraction of light affects the reconstruction quality. One solution involves utilizing multiple SLMs to separately modulate the phase and amplitude, although this approach increases the system complexity [102]. The spatial-multiplexing method [103,104,105] is widely used to achieve full-color holography and effectively utilize the spatial bandwidth product (SBP) of the SLM. Alternatively, time-multiplexing techniques [106] can reconstruct full-color images using one SLM based on the principles of human eye retention. Yet holographic displays for 3D HUDs still have limitations, such as the small space bandwidth product, speckle noise, zeroth-order light elimination, and video-rate CGH algorithm.

## 5. Conclusions and Outlook

In this review, we began with the investigation of PGU modules and compared the disparities among various schemes. Building upon this foundation, we further introduced diverse single-plane AR-HUD systems’ performance while addressing critical challenges pertaining to distortion, dispersion, and poor image contrast ratios. Furthermore, we closely monitored cutting-edge advancements in multi-plane AR-HUDs, AR-HUDs with variable VIDs, and 3D-HUDs. A summary of the optical performance of HUD systems is given in Table 3. Although extensive efforts have been made to improve the VID and FOV, there is still much room for improvement. As the hardware entrance to the automotive metaverse, we expect a full-windshield 3D-HUD that can break current technical bottlenecks.

Aside from the VID, FOV, and system volume, which we extensively examined in this review, other critical requirements of HUDs include a high brightness (greater than 10,000 nit) and high contrast ratio. However, the reflectivity of a windshield is generally about 25%. The most intuitive way to increase brightness is to increase the irradiance from the PGU, but this is bound to thermal management and energy consumption problems. Another possible solution is to change the spectral reflectance of the windshield with multi-layer coatings. However, this solution has critical requirements in coating uniformity. Complex structures based on dielectric resonant gratings can also control the reflectivity of specific wavelengths, but their angular tolerances are small and incompatible with HUDs [108]. Nanoparticle-array-embedded combinators have been proposed based on bipolar localized surface plasmon resonance (LSPR) [109]. High reflectivity is obtained at specific wavelengths while maintaining clear transparency. A study suggested that correlated disordered arrangement attenuates diffraction effects, leaving the background landscape visually unchanged. Aside from brightness, a high ambient contrast ratio is favorable in AR systems for the readability of virtuality. A study proposed by Shin-Tson Wu involved utilizing electronically controlled transmittance with LC film to regulate ambient brightness and combined a DBEF to augment the display brightness [110].

A comprehensive evaluation of AR-HUDs encompassing not only optical architecture but also real-time 3D imaging and meticulous consideration of information processing algorithms are essential. In practice, the utilization of LiDAR sensors has become imperative for AR-HUDs to accurately perceive real-time road conditions while driving. It is worth mentioning that inappropriate or overly complex virtual images can indeed divert a driver’s attention. Therefore, more research should be conducted to rationalize displayed information, enhance interface clarity and conciseness, and develop more user-friendly and intuitive human–machine interfaces in order to reduce attention load. Furthermore, the utmost attention should be given to enhancing user-centric personalized experiences and implementing robust data privacy protection measures.

## Figures and Tables

**Figure 1 micromachines-15-00442-f001:**
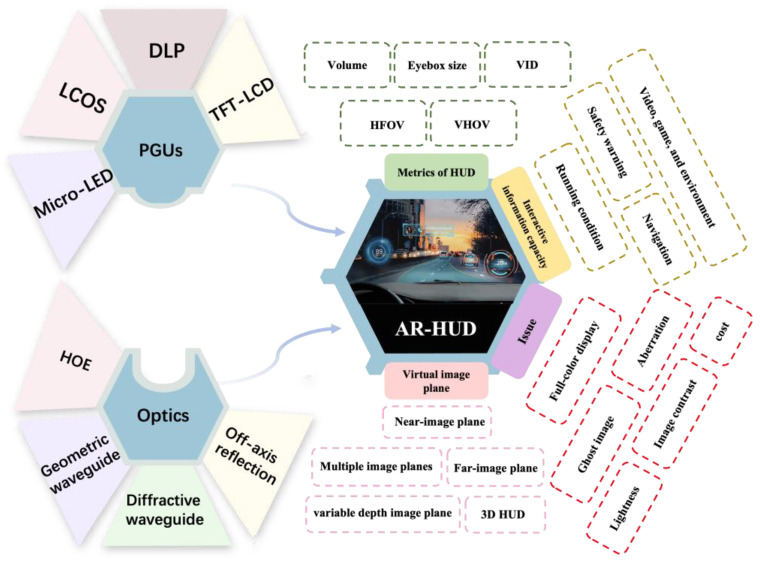
Overview of AR-HUDs.

**Figure 2 micromachines-15-00442-f002:**
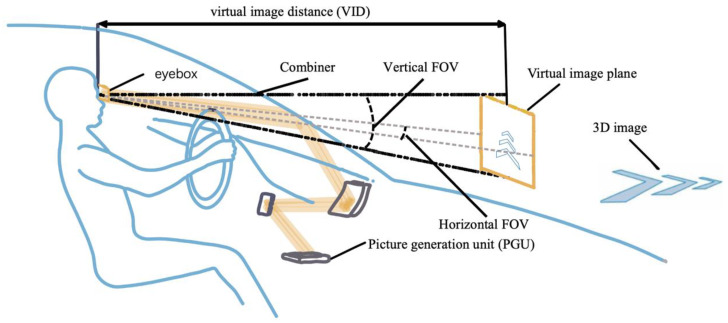
Key metrics of HUDs.

**Figure 3 micromachines-15-00442-f003:**
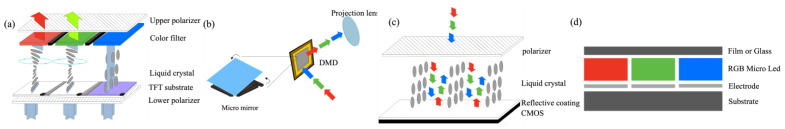
Schematics of typical projectors. (**a**) Transmission LCD-TFT in bright (R and G) and dark (B) states. (**b**) Reflective DLP technology for field sequential color (RGB) operation. (**c**) Reflective LCOS for field sequential color (RGB) operation. (**d**) RGB micro-LED.

**Figure 4 micromachines-15-00442-f004:**
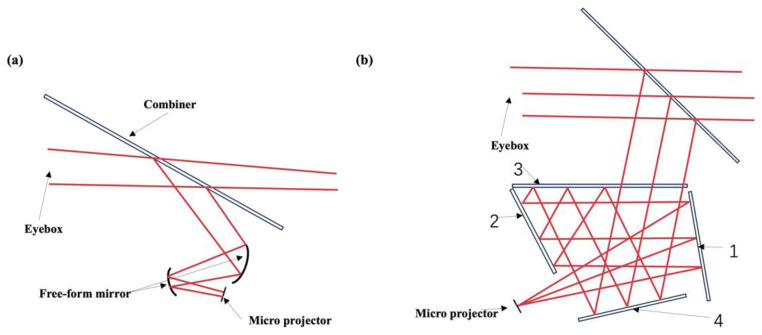
Schematics of a free-space HUD system. (**a**) Schematics of an off-axis reflective HUD with dual free-form mirrors. (**b**) Schematics of a four-mirror off-axis reflective HUD.

**Figure 5 micromachines-15-00442-f005:**
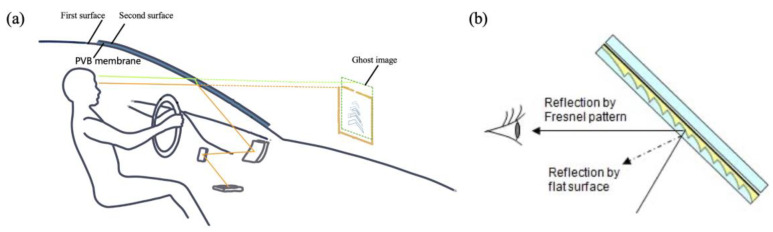
Issue of ghost images in an optical combiner. (**a**) Causes of ghost images. (**b**) A diagram of the elimination of ghost images with the Fresnel pattern [48]. Copyright of IEEE.

**Figure 6 micromachines-15-00442-f006:**
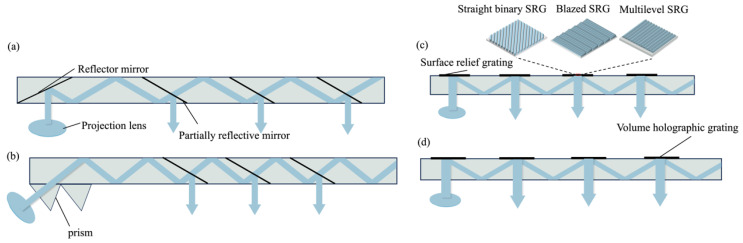
Schematics of geometric and diffractive waveguides. (**a**) A geometric waveguide with mirrors coupled in and partially reflective mirrors coupled out. (**b**) A geometric waveguide with a prism coupled in and partially reflective mirrors coupled out. (**c**) A diffraction waveguide coupled in and out with SRGs (mainly including straight binary grating, blazed grating, and multilevel grating). (**d**) A diffraction waveguide coupled in and out with VHGs.

**Figure 7 micromachines-15-00442-f007:**
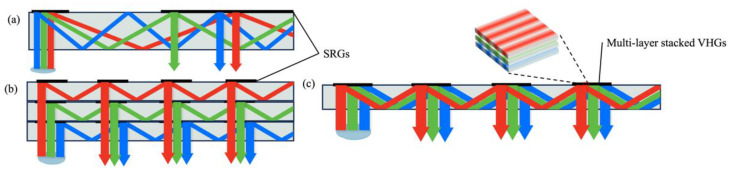
Issue of diffraction dispersion in a diffractive waveguide. (**a**) Schematics of diffractive dispersion. (**b**) Schematics of a multilayer SRG waveguide structure. (**c**) Schematics of a multilayer stacked VHG structure.

**Figure 9 micromachines-15-00442-f009:**
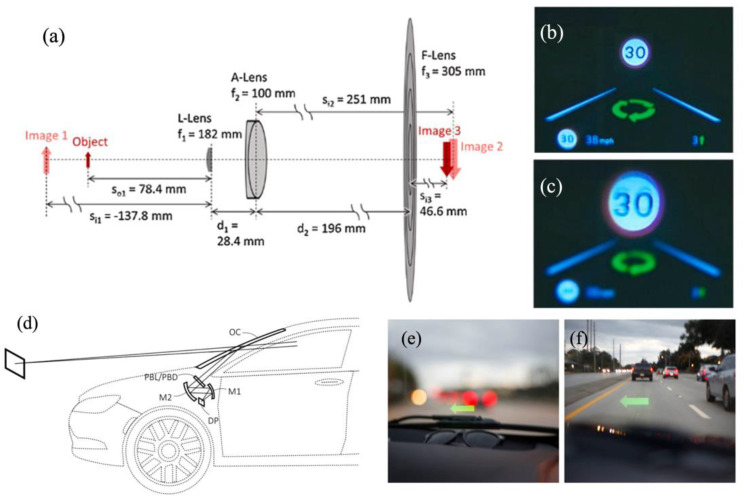
Research progress for variable−depth−image−plane HUD systems. (**a**) Optical architectures achieving the intended image depth and size variations where the L−Lens focal length is set to 182 mm. (**b**) Images captured from the integrated HUD system with the camera focused on the speed−limit sign as it moves from 1.95 to 1.52 m in the distance, which corresponds to the L−Lens powers of 5.5 and (**c**) −4.5 diopters, respectively [78], copyright of Elsevier. (**d**) Schematics of the optics in an off-axis reflective HUD. (**e**,**f**) Photograph captured through an HUD focusing at (left) short and (right) long virtual image distances enabled by passively driven Pancharatnam−Berry lenses (PBLs) [83], copyright of John Wiley and Sons.

**Figure 10 micromachines-15-00442-f010:**
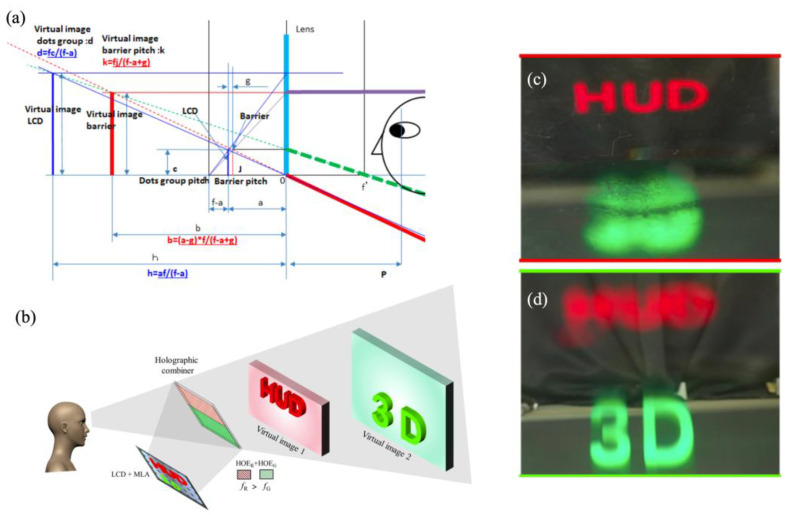
Research progress for 3D HUD systems. (**a**) Schematics of a 3D AR-HUD system based on a parallax barrier and eye tracking [85], copyright of John Wiley and Sons. (**b**) Schematics of a 3D AR-HUD system integrating a microlens-array-based 3D display module with a holographic combiner. (**c**,**d**) Results when the camera is focused on the depth plane corresponding to red and green [87], copyright of Optica.

**Table 1 micromachines-15-00442-t001:** HUD parameters at different stages of development.

Metrics	C-HUD [23]	W-HUD [24]	AR-HUD [25]	Ideal
FOV	<5° × 1.4°	6° × 2°	>13° × 5°	>20° × 10°
VID	~1 m	<4.5 m	>7 m	>20 m
Brightness	<10,000 cd/m^2^	<10,000 cd/m^2^	>10,000 cd/m^2^	>10,000 cd/m^2^
Volume	<2 L	<4 L	<10 L	<3 L

**Table 2 micromachines-15-00442-t002:** Critical parameters of PGUs.

Micro Projector	Principle	Resolution	Optical Efficiency	Image Contrast	Formfactor
LCD-TFT	Transmissive	Low	Low	Medium	Small
DLP	Reflective	Medium	High	High	High
LCOS	Reflective	High	Medium	Medium	Medium
Micro LED	Self-emissive	High	High	High	Small

**Table 3 micromachines-15-00442-t003:** Critical parameters of AR-HUDs.

	Technology	FOV	VID	Eyebox	Volume
Near-image plane	Off-axis four-mirror system [42]	6° × 3°	5 m	106 × 66 mm	big
Far-image plane	Geometric waveguide [54]	24° × 15°	>10 m	80 × 80 mm	small
	SRG [107]	16° × 14.25°	>10 m	50 × 50 mm	~2 V
	VHG [58]	24° × 12.6°	>10 m	50 × 100 mm	small
Multiple image planes	Dual optical routes [72]	6° × 2°/10° × 3°	2/8~24 m	120 × 60 mm	—
	Laser scanning [73]	6° × 1°/8° × 2°	2/5 m	130 × 40 mm	—
Variable-depth image plane	Liquid lens [78]	—	2 m	86 × 84/139 × 139 mm	—
	Spatial light modulator [84]	—	3–30 m	—	—
3D-HUD	Parallax barrier [85]	—	3.5 m	126 × — mm	—
	Binocular parallax based on a lens array	20° × 7°	—	130 × 60 mm	—
	Microlens-array-based binocular parallax [86]	—	2.2 m	—	—
	Computer-generated hologram [99]	5° × 3°	1.9–4.5 m	—	—

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
