# Peer review of "Automotive Augmented Reality Head-Up Displays"

_micromachines, 2024, doi:10.3390/mi15040442_

Round 1

Reviewer 1 Report

Comments and Suggestions for Authors

The paper reviews some key metrics of HUD, investigates the structure of various PGUs, and focuses on the development status of AR-HUD, analyzes the advantages and disadvantages of existing technologies. The paper mentions that PGUs uses TFT-LCD, DLP, or LCOS to generate images, but there is currently another technology that uses Micro-LED to generate images, which is also a hot research topic. Therefore, it is recommended that the author supplement some information on the current development status of Micro LED as PGUs, as well as their prospects for future HUD applications.

Reviewer 2 Report

Comments and Suggestions for Authors

This article has been carefully written, and I believe it is worth publishing.

However, it is worth mentioning in this work that, sometimes, the augmented reality graphics may direct the driver's attention away from critical road elements.
